# Illness prevalence and symptoms in youth floorball players: a one-season prospective cohort study involving 471 players

Nirmala Kanthi Panagodage Perera ![ORCID],[1,2] Markus Waldén ![ORCID],[1,3] Hanna Lindblom ![ORCID],[1] Ida Åkerlund ![ORCID],[1] Sofi Sonesson ![ORCID],[1] Martin Hägglund ![ORCID] [1]

[1]Department of Health, Medicine and Caring Sciences, Unit of Physiotherapy, Sport Without Injury ProgrammE (SWIPE), Linköping University, Linköping, Östergötland, Sweden
[2]Australian Institute of Sport, Burce, Australian Capital Territory, Australia
[3]Department of Orthopaedics, Hässleholm-Kristianstad Hospitals, Hässleholm, Sweden

**Correspondence to**
Professor Martin Hägglund;
martin.hagglund@liu.se

## ABSTRACT

**Objective** To describe weekly illness prevalence and illness symptoms by sex in youth floorball players during one season.

**Design** Prospective cohort study.

**Setting** Players who were registered to play community level floorball during the 2017–2018 season (26 weeks) in two provinces in southern Sweden.

**Participants** 471 youth players aged 12–17 years. Mean (SD) age for 329 male players 13.3 (1.0) years and 142 female players 13.7 (1.5) years.

**Primary and secondary outcome measures** Weekly self-reported illness prevalence and illness symptoms according to the 2020 International Olympic Committee's consensus recommendations.

**Results** 61% of youth floorball players reported at least one illness week during the season, with an average weekly illness prevalence of 12% (95% CI 10.8% to 12.3%). The prevalence was slightly higher among females (13%, 95% CI 11.6% to 14.3%) than males (11%, 95% CI 9.9% to 11.7%), prevalence rate ratio 1.20 (95% CI 1.05 to 1.37, p=0.009). In total, 49% (53% male, 43% female) of illness reports indicated that the player could not participate in floorball (time loss), with a mean (SD) absence of 2.0 (1.7) days per illness week. Fever (30%), sore throat (16%) and cough (14%) were the most common symptoms. Female players more often reported difficulty in breathing/tight airways and fainting, and male players more often reported coughing, feeling tired/ feverish and headache. Illness prevalence was highest in the peak winter months (late January/February) reaching 15%–18% during this period.

**Conclusions** Our novel findings of the illness prevalence and symptoms in youth floorball may help direct prevention strategies. Athletes, coaches, parents and support personnel need to be educated about risk mitigation strategies.

**Trial registration number** NCT03309904.

## Strengths and limitations of this study

► The first study to evaluate self-reported prevalence, seasonal variations and illness symptom clusters by sex according to the 2020 International Olympic Committee's methods for recording and reporting epidemiological data consensus recommendations.
► Illness data from a large sample of male and female recreational youth floorball players over a 26-week period.
► Self-reported illness data may be subject to recall and interpretation bias, and the actual diagnosis was not available.
► Injury was the primary outcome of interest in the intervention study; players who reported an injury in the weekly survey could not report a concurrent illness during the same week, and illness data thus represent minimum illness prevalence rates in the study cohort.

Strenuous exercise during acute infection can increase the risk of serious medical complications, including sudden death.[2 5–7] Performance impairments from upper respiratory tract illness may also last for 2–4 days after complete recovery.[2] Infectious illness can reduce muscle strength and impair motor coordination,[7] decrease aerobic and endurance capacity[5] and alter metabolic function[7] during and/or after the infection.

In elite youth sports, illness prevalence is higher among athletes who participate in endurance sports (23%) than those who participate in technical (10%) or team (8%) sports.[8] In elite adult sports, illness is more common among female athletes and those who participate in outdoor winter sports[9]; however, this relationship is unclear in elite youth sports.[10–13] The incidence of illness among recreational youth athletes during a season or competition has not been well studied. There is a need for increased knowledge about this population, given: (1) the

## INTRODUCTION

Acute illness presents a significant health burden to athletes, interfering with their training, reducing sport performance and causing time loss from competitions,[1–3] and may increase the risk of subsequent injuries.[4]

scarcity of available data, (2) that most youth athletes participate in sport at recreational level, (3) it may not be appropriate to extrapolate the results from adult and elite youth cohorts to this population owing to the differences in training load (intensity and volume), competition pressure, access to resources and travelling and (4) to provide direction for illness prevention initiatives. In this context, this study aims to be the first to: (1) describe the epidemiological profile of weekly illness prevalence by sex and (2) describe self-reported illness symptoms by sex among youth floorball players during a full 26-week season.

## METHODS

This prospective cohort study was part of a Sport Without Injury ProgrammE (SWIPE) cluster randomised controlled trial (RCT) in floorball evaluating the efficacy of the *Knee Control* injury prevention exercise programme (Clinical trials registration NCT03309904). Representatives from the Swedish Floorball Federation informed the development of the *Knee Control* injury prevention exercise programme, and coaches (end-users) were involved in the RCT. Written informed consent was collected from all participating players, and from legal guardians for players below 15 years of age. Since the 15-min warm-up training intervention was not believed to influence illness occurrences, we included all players in both the intervention and control group arms of the RCT for analysis in this substudy. Youth male and female community-level floorball players aged 12–17 years from two provinces in southern Sweden (Östergötland and Småland) registered to play floorball in a club during the 2017–2018 season (26 weeks) were included.[14 15] Floorball is a winter sport played indoors on a playing surface made of rubber or parquet, using a plastic stick and light plastic ball on a 40×20 m pitch. There are five field players and one goalkeeper on the court in each team with free substitutes.

To prospectively collect illness data, all players completed a weekly web-based survey during the season using the Oslo Sports Trauma Research Center (OSTRC) questionnaire on health problems.[16] If players answered anything other than the first option on the OSTRC questionnaire question 1, that is, 'full participation in floorball training and matches during the past week without health problems', they were asked to indicate whether their health problem was an injury/complaint or illness. Players who selected illness then answered a question about their main illness symptoms during the past week, and which day(s), if any, they had missed scheduled floorball training or matches due to illness (box 1). Injury was the primary outcome of interest in the intervention study, and if a player reported an injury in the weekly survey they could not report a concurrent illness during the same week. Weekly surveys were administered each Sunday at 18:00 hours via a web-link sent via short messaging service (SMS) and email to the players, using the online survey software esMakerNX3 V.3. If the players

> **Box 1  Weekly survey questions used to collect illness data during the 26-week season**
>
> Have you had any difficulties participating in normal floorball training during the past week due to health problems (such as pain, aches, tenderness, stiffness), injury or illness?*
> ► Full participation without health problems, injury or illness
> ► Full participation, but with health problems, injury or illness
> ► Reduced participation due to health problems, injury or illness
> ► Could not participate due to health problems, injury or illness
> Was your current health problem an injury/complaint or illness?
> ► Injury
> ► Illness
> Select the option that best matches your major illness symptoms. (Enter using the drop-down list)
> ► Fever
> ► Feeling tired/feverish
> ► Swollen glands
> ► Sore throat
> ► Nasal congestion/runny nose/sneezing
> ► Coughing
> ► Difficulty breathing/tight airways
> ► Headache
> ► Nausea
> ► Vomiting
> ► Diarrhoea
> ► Constipation
> ► Fainting
> ► Skin rash/itchy skin lesions
> ► Irregular or abnormal heartbeats, chest pain or discomfort with exercise
> ► Abdominal pain
> ► Other pain
> ► Numbness
> ► Nerve tingling/pain
> ► Anxiety
> ► Depression/feeling down
> ► Irritation
> ► Eye symptoms
> ► Ear symptoms
> ► Urogenital/gynaecological symptoms
> ► Other
> Which day(s) during the past week did you miss scheduled floorball training or matches due to illness?
>
> *Question from the Oslo Sports Trauma Research Center questionnaire on health problems[16]

did not complete the survey on the same or the following day, two additional reminders were sent via SMS on the Tuesday and Thursday. All player reports were checked by a research coordinator (physiotherapist) for completeness and accuracy, and this coordinator contacted the player in case of any suspected errors in the registration. Player illness data were included for the full season or up until censoring, for example, due to quitting floorball, or a season-ending injury.

A total of 7503 weekly player reports were registered over the 26 weeks; 745 weekly reports were excluded because the player was absent from floorball for the whole week due to reasons other than injury or illness

(eg, holiday). The final analysis thus comprised 6758 weekly player reports. The average weekly illness prevalence for the whole season was calculated by dividing the number of weeks a player had reported illness by the total number of weekly reports. Illness prevalence was also calculated for each week of the season by dividing the number of players reporting an illness by the number of questionnaire responses for that week. If a player reported multiple illness episodes, these were included as separate events. Illnesses present at the start of the season were included in prevalence calculations. According to the 2020 International Olympic Committee's consensus recommendations, we grouped illness symptoms into clusters post study.[17] Sex differences in average weekly illness prevalence were compared using a prevalence rate ratio (PRR) and corresponding 95% CI. The frequency of reported illness symptoms was compared between sexes using the $\chi^2$ test, treating each athlete's weekly illness reports as independent observations, and mean time loss (number of days absence) per illness week using the Student's t-test. Data were analysed using SPSS V.25.0 (IBM SPSS Statistics 2015). The significance level was set to p<0.05.

### Patient and public involvement

Patients and/or the public were involved in the design, or conduct, or reporting, or dissemination plans of this research.

### RESULTS

There were 471 players included in 47 teams. The mean age for the 329 male and 142 female players was 13.3 (SD 1.0) and 13.7 (SD 1.5) years, respectively. The mean number of weekly reports per player was 14.4 (SD 7.8), with 13.5 (SD 8.1) for males and 16.3 (SD 6.8) for females. In total, 61% of players (n=288, 60% males vs 64% females) reported at least one illness during the season. Of the total number of 6699 weekly reports with complete data for OSTRC question 1 (data were missing for 59 reports), 74% (76% male vs 71% female) indicated that players were able to fully participate in floorball without health problems, in 14% (n=948) the player reported an injury (and hence could not report an illness), and in 12% (n=781) of all reports the player reported an illness. Among the 781 weeks with a reported illness, the player reported full floorball participation in 196 weeks (25%; 21% males vs 31% females), reduced participation in 202 weeks (26%; 26% for both males and females), and inability to participate in floorball due to illness in 383 weeks (49%; 53% males vs 43% females). The mean (SD) time loss from training and matches for all illness weeks was 2.0 (1.7) days, with no difference between sexes (male 2.0 (1.6) vs female 1.9 (1.7) days, p=0.340).

### Illness prevalence during the season

The average weekly illness prevalence for the whole season was 12% (95% CI 10.8% to 12.3%) for all players,

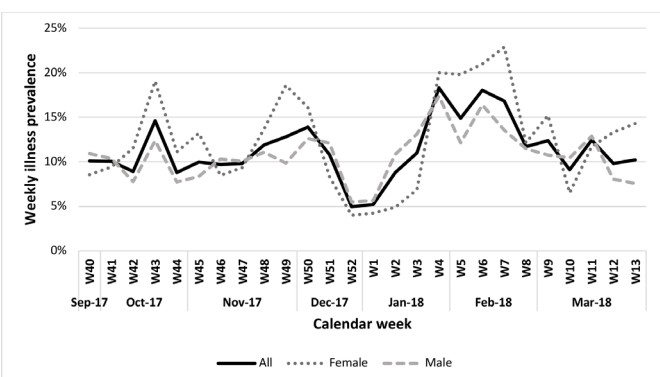

**Figure 1** Weekly illness prevalence for all players and by sex during the 26-week floorball season.

with a slightly higher prevalence among females (13%, 95% CI 11.6% to 14.3%) than males (11%, 95% CI 9.9% to 11.7%), PRR 1.20 (95% CI 1.05 to 1.37, p=0.009). Illness prevalence ranged from 5% to 25% in different teams.

The weekly illness prevalence ranged from 5% to 18% over the 26 weeks (4%–23% for female vs 5%–17% for male players). Illness prevalence was highest in calendar weeks 4 to 7 (peak winter weeks in January/February) and ranged from 15% to 18% during this period (20%–23% for females vs 12%–17% for males) (figure 1).

### Illness symptoms

The most common self-reported illness symptoms were fever (30%), sore throat (16%) and cough (14%). Female players more often reported difficulty in breathing/tight airways and fainting, and male players more frequently reported coughing, feeling tired/feverish and headache (table 1).

Weekly symptom prevalence for non-specific illness symptoms (fever, feeling tired/feverish) and upper/lower respiratory symptoms (sore throat, coughing, nasal congestion/runny nose/sneezing, difficulty breathing/tight airways) were high during January and February (weeks 4–7). Respiratory symptoms also peaked in mid-October (figure 2).

### DISCUSSION

The main findings of this study were that: (1) 61% of youth floorball players reported at least one illness week during the season, with an average weekly illness prevalence of 12%, (2) 49% of all weekly reports with an illness indicated that the player was not able to participate in floorball (time loss), (3) illness prevalence was slightly higher among females, (4) fever (30%) and upper and lower respiratory symptoms such as sore throat (16%), cough (14%) and nasal symptoms (13%) were the most common and (5) illness prevalence was highest in the peak winter months (late January/February), affecting 15%–18% of the players during this period.

**Table 1** Self-reported illness symptoms reported during the 26-week floorball season sorted by illness symptom cluster

| Illness symptom clusters | All | Female | Male | P value female versus male | Illness impact on participation in floorball | | |
|---|---|---|---|---|---|---|---|
| | | | | | Full participation but with illness | Reduced participation due to illness | Could not participate due to illness |
| Upper/lower respiratory (total) | 387 (49.9) | 156 (52.7) | 231 (48.1) | 0.215 | 133 | 104 | 150 |
| Sore throat | 122 (15.7) | 46 (15.5) | 76 (15.8) | 0.913 | 22 | 42 | 58 |
| Coughing | 105 (13.5) | 30 (10.1) | 75 (15.6) | 0.030 | 36 | 23 | 46 |
| Nasal congestion/runny nose/sneezing | 100 (12.9) | 32 (10.8) | 68 (14.2) | 0.175 | 30 | 29 | 41 |
| Difficulty breathing/tight airways | 60 (7.7) | 48 (16.2) | 12 (2.5) | <0.001 | 45 | 10 | 5 |
| Non-specific illness (total) | 278 (35.8) | 96 (32.5) | 182 (37.9) | 0.122 | 19 | 72 | 187 |
| Fever | 230 (29.6) | 86 (29.1) | 144 (30.0) | 0.779 | 9 | 48 | 173 |
| Feeling tired/feverish | 42 (5.4) | 8 (2.7) | 34 (7.1) | 0.009 | 9 | 23 | 10 |
| Other pain | 6 (0.8) | 2 (0.7) | 4 (0.8) | NC | 1 | 1 | 4 |
| Neurological—headache | 32 (4.1) | 4 (1.4) | 28 (5.9) | 0.002 | 12 | 10 | 10 |
| Gastrointestinal (total) | 25 (3.3) | 8 (2.8) | 17 (3.5) | 0.520 | 4 | 10 | 11 |
| Nausea | 14 (1.8) | 4 (1.4) | 10 (2.1) | 0.457 | 2 | 6 | 6 |
| Vomiting | 6 (0.8) | 2 (0.7) | 4 (0.8) | NC | – | 1 | 5 |
| Abdominal pain | 3 (0.4) | 2 (0.7) | 1 (0.2) | NC | 2 | 1 | – |
| Diarrhoea | 2 (0.3) | – | 2 (0.4) | NC | – | 2 | – |
| Cardiovascular—fainting | 20 (2.6) | 20 (6.8) | – | NC | 17 | 1 | 2 |
| Other—unspecified | 27 (3.5) | 8 (2.7) | 19 (4.0) | 0.354 | 6 | 5 | 16 |
| Swollen glands | 10 (1.3) | 2 (0.7) | 8 (1.7) | NC | 2 | – | 8 |
| Otological—ear symptoms | 3 (0.4) | 2 (0.7) | 1 (0.2) | NC | 1 | – | 2 |
| Psychological (total) | 3 (0.4) | 2 (0.7) | 1 (0.2) | NC | 1 | – | 2 |
| Anxiety | 2 (0.3) | 1 (0.3) | 1 (0.2) | NC | 1 | – | 1 |
| Irritation | 1 (0.1) | 1 (0.3) | – | NC | – | – | 1 |
| Ophthalmological—eye symptoms | 1 (0.1) | – | 1 (0.2) | NC | 1 | – | – |
| Total | 776* (100) | 296 (100) | 480 (100) | | 194 | 202 | 380 |

Values are number of weekly reports (% of total weekly reports) with each reported symptom.

*Illness symptoms missing for five weekly illness reports. P values for comparison of illness symptom frequency between sexes using the $\chi^2$ test (not calculated (NC) if expected value in any cell is less than 5).

## Illness prevalence and symptoms

The average weekly illness prevalence was 12%, which is comparable to Norwegian youth elite athletes participating in technical sports (10%),[8] but higher than for Norwegian elite youth athletes participating in team sports (8%–9%),[8 18] Dutch junior tennis players (6%),[10] and Irish elite youth track and field athletes (7%).[19] The Swedish floorball season runs from September to March (autumn and winter months) when illness is more prevalent compared with summer months when athletics and tennis are played. Therefore, youth players are likely to be exposed to different viruses in school or public places. Also, some established norms associated with team sports, ranging from sharing drinking bottles, towels, equipment and locker rooms, to social behaviour such as hugging, shaking hands and celebrations can enhance transmission of air-borne and droplet-borne pathogens like influenza.

Sports clubs can implement various behavioural, lifestyle and medical intervention strategies to ensure that sporting environments are safe spaces for sports participation, and this has become even more crucial during the current COVID-19 pandemic.[9 20 21] The education of athletes, coaches, parents and support personnel about risk-mitigation strategies for viral infections is important. This includes physical distancing to reduce the risk of transmission, cough and sneeze etiquette, wearing masks

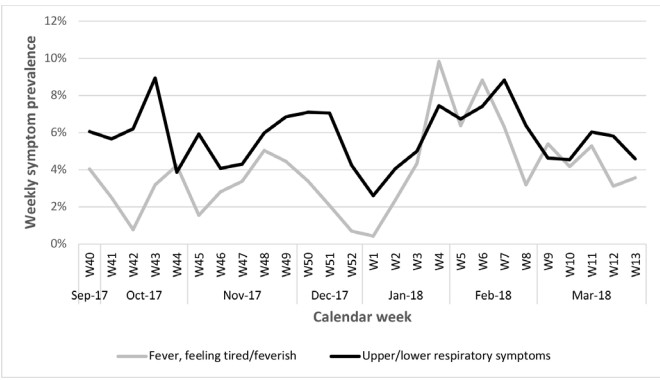

**Figure 2** Weekly symptom prevalence for all players during the 26-week floorball season.

when symptomatic (coughing, sneezing) to reduce the transmission of potentially infectious respiratory droplets, regular hand hygiene, not sharing equipment such as drinking bottles and strategies that facilitate good quality sleep and sleep hygiene practices.[9] Improved health literacy, including awareness of self-monitoring of respiratory symptoms (even if mild), could help to reduce the risk of spreading the infection to team mates.[20] Medical strategies also include implementing illness prevention guidelines, illness surveillance and respiratory symptom screening, identifying at-risk and vulnerable individuals and implementing full preventative precautions during high-risk periods.[9]

The most common symptoms were fever (30%), and upper/lower respiratory symptoms such as sore throat (16%), cough (14%) and nasal symptoms (13%). These symptoms peaked during the winter season in late January/February (weeks 4–7) in our floorball cohort, which was concurrent with a sharp increase in influenza B/Yamagata in the general population of young people in Sweden during week 3, with a peak in week 7.[22] This indicates that influenza could be one common cause of illness in our Swedish youth floorball cohort.

Our data show that one in every two reported illnesses led to complete time loss from the sport, with a mean absence of 2 days per illness week. Participating in sports during febrile illness can worsen the illness or lead to other detrimental outcomes and return to activity is only recommended with the resolution of fever.[23] Therefore it is worrisome that 1 in 4 athletes who reported fever as the main symptom, and 3 of 4 athletes who reported feeling tired/feverish, also reported full or partial participation in floorball.

### Comparison between sexes
We observed a slightly higher weekly illness prevalence in female youth floorball players than males (13% vs 11%), and our findings concur with the results from several different sports in Norwegian youth elite athletes.[8 18] In elite adult sports as well, female athletes appear to be more prone to illness,[9] yet the influence of sex on illness prevalence remains unclear in elite youth sports.[10–13] Overall, reported symptom clusters were similar between sexes,

while among upper/lower respiratory symptoms more female players (16% vs 3% males) reported difficulty in breathing/tight airways, while males more commonly reported coughing (16% vs 10% females). Furthermore, approximately 1 in 20 female players reported fainting, yet no male players did so. The reason for this sex discrepancy is unclear.

### Methodological considerations
Self-reported illness data rely on young players' memory, and consequently are subject to recall bias. The use of prospective weekly reports limits the recall period to 1 week. Additionally, self-reported illness reports are open to interpretation bias of the survey question, and it is possible that young people may not fully understand illness-related questions. Question responses were standardised by providing a list of symptoms to collect self-reported data as accurately as possible. Since the diagnosis was not available from self-reported illness, we followed the 2020 International Olympic Committee's methods for reporting illness data by symptoms, and symptom clusters.[17] The study was limited by a low overall survey response rate (61%), which is similar to two previous studies in Swedish and Norwegian youth elite athletes (response rates 60%–66%).[8 24] Female players had a higher survey response rate than males (16 vs 14 weekly reports during the season). Still, our sample of female floorball players was smaller than that of male players and data may thus be less robust. It is unknown whether there is a bias towards higher or lower response rates in weeks when the athlete had experienced a health problem, thus risking either overinflation or underinflation of the illness prevalence rate. Because of the way in which the survey was structured, there is a possibility that illness data might be under-reported. Injury was the primary outcome of interest in the intervention study, and if a player reported an injury in the weekly survey they could not report a concurrent illness in the same week. Therefore, our illness prevalence numbers represent the minimum illness rates in the study cohort. If we assume the same illness prevalence during injury weeks as non-injury weeks the 'true' weekly illness prevalence in our cohort would be slightly higher at 13% (95% CI 12.4% to 14.0%).

### CONCLUSIONS
This prospective study showed that 61% of youth floorball players reported at least one illness week during the season, with an average weekly illness prevalence of 12%. Our novel findings relating to the prevalence and pattern of illness in youth floorball may help to direct prevention strategies. Athletes, coaches, parents and support personnel need to be educated about risk mitigation strategies for common infections. Improved awareness about the self-monitoring of respiratory symptoms could help to reduce the risk of spreading the infection to team mates.

**Acknowledgements** The authors would like to thank Taru Tervo, PhD, School of Sport Sciences, Umeå University, and Emil Risberg, of The Swedish Floorball Federation, for input on the study plan, surveys and research questions, as well as administrative assistance. We also thank the participating clubs, coaches and players for their participation in the study. The Sport Without Injury ProgrammE (SWIPE) was established at Linköping University, Linköping, Sweden, through grants from the Swedish Research Council and the Swedish National Centre for Research in Sports.

**Contributors** NKPP analysed and interpreted the data and produced the first draft of the manuscript. MW, HL, IÅ and SS provided critical comments on the manuscript. MH conceptualised the study, interpreted the data and provided critical comments on the manuscript. All authors were involved in designing the study and approved the final draft for submission. MH is responsible for the overall content as the guarantor.

**Funding** The Sport Without Injury ProgrammE is funded by the Swedish Research Council (2015-02414 and 2018-03135) and the Swedish Research Council for Sport Science (P2018-0167).

**Competing interests** None declared.

**Patient consent for publication** Not applicable.

**Ethics approval** The study was conducted in line with the Declaration of Helsinki, and approved by the Swedish Ethical Review Authority (Dnr 2017/294-31).

**Provenance and peer review** Not commissioned; externally peer reviewed.

**Data availability statement** Data are available upon reasonable request. Deidentified data may be shared upon reasonable request.

**ORCID iDs**
Nirmala Kanthi Panagodage Perera http://orcid.org/0000-0001-6110-8945
Markus Waldén http://orcid.org/0000-0002-6790-4042
Hanna Lindblom http://orcid.org/0000-0002-1533-6872
Ida Åkerlund http://orcid.org/0000-0002-0338-3647
Sofi Sonesson http://orcid.org/0000-0001-8670-5666
Martin Hägglund http://orcid.org/0000-0002-6883-1471

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
