## [Reviewer comments · BMJ Open]

ARTICLE DETAILS

TITLE (PROVISIONAL)	Illness prevalence and symptoms in youth floorball players – a one-season prospective cohort study involving 471 players.
AUTHORS	Panagodage Perera, Nirmala; Waldén, Markus; Lindblom, Hanna; Åkerlund, Ida; Sonesson, Sofi; Hagglund, Martin

VERSION 1 – REVIEW

REVIEWER	Philippe Tscholl F-MARC
REVIEW RETURNED	24-May-2021

GENERAL COMMENTS	Congratulations to the authors to this very nice manuscript. It is very well written, sound and new. Although it would have been more valuable to have the injury and illness reports in the same manuscript, the methods of the manuscript are important to be published. The results are less surprising, and probably not due to sporting activity in floorball however represent the incidence of illnesses in the normal population of adolescents in Sweden. Since the author conclude, that prevention programs should be designed, and parents, coaches and athletes educated, it would be interesting to see the differences of the 2 different provinces (without mentioning their names of course) or even the different teams, whether health measures should be taken into considerations. Unfortunately, the fact that illnesses could not be recorded during injury is a major limitation and is not logic.
--

REVIEWER	Nicola Sewry University of Cape Town, Human Biology
REVIEW RETURNED	26-Jul-2021

GENERAL COMMENTS	Dear Authors, thank you for this opportunity to review your manuscript. I have a few general comments: A large piece of missing data in illness surveillance is the time-loss aspect. You mention having these data and touch on it briefly, but I think if you were to add detail on which illnesses led to time-loss, classify your time-loss illnesses into the varying categories (eg. 1 day, 1 week etc) this will make for a much stronger paper. Specific Comments: Abstract: Your title and objective in the abstract do not match with regards to your main objectives?
---

Please include the stats into your outcome measures
In the results, fever should not be included in the upper/lower cluster.
Conclusions, "Flu-like symptoms dominated." this is not a full sentence, and you will see my comments to come regarding the term "flu-like".

Overall comment: I would recommend you have an English editor read through your work, as there are many grammar errors. I have not listed them.

Introduction:

There is no mention of floorball in the introduction. There needs to be some context of floorball, and the age group of these athletes (some were only 12 years old), and not elite. Is floorball indoor or outdoor? Is it winter or summer? If it's indoors, is there ventilation, is it on ice? These environmental conditions should be explained to the reader as, I had not heard of floorball before this paper.

Methods:

Please include a statement regarding informed consent and assent from the parents and children (I assume this was done seeing as this trial was such a large one and you have ethics), but please make sure you explicitly write this.

Seeing as an athlete could not report an illness if an injury was reported, could you please indicate the number of weeks this was the case (eg. in 200 player weeks an injury was reported and therefore an illness could not be reported due to methodological errors)?

Could you please explain why you did not use illness incidence? per player days... or player weeks? This is what is recommended in the consensus statement.

Also, considering that many players had multiple illnesses, consider reporting whether the illnesses were subsequent illness, recurrent or subsequent local illnesses. This would be appropriate seeing as you are attempting to follow the consensus guidelines.

Statistical Analysis:

What was your level of significance?

An assumption of the Chi-square test is that you compare frequencies and not percentages? What did you do, seeing you write that you used proportions? Please check this.

Results:

Line 139-142: as mentioned in the earlier comments, here you give some information regarding the time-loss data, it would be of interest to the clinician and researcher to give more detail on these data.

Include the types of illnesses which result in this, are females more or less affected than males? Is this different during the season?

Illness symptoms: do you have the duration of symptoms? this could add more value than just what symptoms each player had?

Table 2: You appear to have compared the sexes (you report a p-value in the narrative), the p-values should be included in this table, otherwise it appears as if you are hiding data.

Please check your % of non-specific illnesses for all.

Swollen glands: this only belongs in upper/lower if they are neck glands, do you have in the information to prove they were neck glands? if not they should be removed.

Line 172: You have not classified these as flu-like, you shouldn't be writing the narrative like this then. Same goes for figure 2.

	Discussion: You list the key findings, however you do not discuss finding 2. Please add this into your discussion. Line 184: again you include fever in upper/lower cluster, this goes against your original groupings. Line 197: again, you mention 1 in 2 illnesses lead to time-loss, this needs to be further investigated in the results please. Line 201: Please rather use the terminology of the virus (SARS-CoV-2), not the "Novel Coronavirus pandemic" Line 201-206: Please revise this sentence, it is much too long. Line 208: "medial"? Line 211-228: Here you have extrapolated your data beyond the scope of your article. You have performed no PCR testing to determine that your illnesses were influenza. Vaccines are beyond the scope of this article. Comparison between sexes: Please remember the average age of your population was 13, at this age females are more mature than males, so the differences between males and females would be smaller. This paragraph reads as though you investigated elite adults. Line 242-244: I am very confused as to what you are trying to say here, but again this seems to apply to adult females anyway, and not 13 year olds. Line 247: please include that these were 13 year old children answering a medical questionnaire, there is a chance that they did not fully understand the illness questions. Line 251-252: This is not a very high response rate, there were 26 weeks, so please include that overall this was a low response rate (58%). Conclusions: Please remove the flu-like sentence. And look at the repetition of "risk mitigation".
--	--

VERSION 1 – AUTHOR RESPONSE

Reviewer: 1

Dr. Philippe Tscholl, F-MARC

Comments to the Author:

Congratulations to the authors to this very nice manuscript. It is very well written, sound and new.

Although it would have been more valuable to have the injury and illness reports in the same manuscript, the methods of the manuscript are important to be published. The results are less surprising, and probably not due to sporting activity in floorball however represent the incidence of illnesses in the normal population of adolescents in Sweden.

Response: Thank you for the positive remarks on our study and manuscript. We agree that illness prevalence in our community sport setting to a large extent represents what is found in the normal adolescent population, not the least since many youths partake in community sport. There are, however, a few sport-specific illness contraction points of interest, e.g. close contact between athletes in-doors and during transportation, sharing the same towels and/or bottles, etc.

Since the author conclude, that prevention programs should be designed, and parents, coaches and athletes educated, it would be interesting to see the differences of the 2 different provinces (without mentioning their names of course) or even the different teams, whether health measures should be

taken into considerations.

Response: The illness prevalence varied between 5% - 25% in the different teams (overall 12%). We have added this information to the manuscript on lines 155-156.

Unfortunately, the fact that illnesses could not be recorded during injury is a major limitation and is not logic.

Response: We agree that this is a limitation, and thus, the illness prevalence reported in our study represent the minimum illness rates in our cohort, as also stated under “Methodological considerations”. The reason for this approach was because the primary purpose of the RCT was to study injury preventive efficacy of a prevention training programme, thus injury being our primary outcome. The response skip logic was used to minimize response burden for the youths, as a repeat of questions related to additional illness occurrence would add further questions to the survey. As noted, the response rate was already somewhat low (61%) and survey fatigue may be one reason for this common problem in studies using athlete self-report of health problems. Importantly, if we assume the same prevalence during injury weeks as during non-injury weeks, the “true” weekly illness prevalence would be only slightly higher; 13% instead of 12%. We believe this is an important clarification and have added the following sentence on lines 254-256: “If we assume the same illness prevalence during injury weeks as non-injury weeks (12%), the “true” weekly illness prevalence in our cohort would be slightly higher at 13% (95% CI 12.4–14.0%).”

Reviewer: 2

Miss Nicola Sewry, University of Cape Town

Comments to the Author:

Dear Authors, thank you for this opportunity to review your manuscript.

I have a few general comments:

A large piece of missing data in illness surveillance is the time-loss aspect. You mention having these data and touch on it briefly, but I think if you were to add detail on which illnesses led to time-loss, classify your time-loss illnesses into the varying categories (eg. 1 day, 1 week etc) this will make for a much stronger paper.

Response: Thank you for the positive remarks on our study and manuscript. We agree time-loss is of interest in relation to illness. Since we rely on athlete self-report of illness data (occurrence and symptoms) we can only report time-loss in relation to weekly reports (prevalence). It is not possible with any certainty to confirm whether e.g. illness reporting over two weeks relate to the same illness or different illness episodes. We have extended and revised Table 2 and added information about illness impact on participation in sport (full participation, reduced participation, could not participate, as per OSTRC questionnaire question 1) for each reported symptom and symptom cluster. We have also added the mean days absence per illness week in total and for each sex in the “Abstract” and in the “Results” line 150-151. There was no difference in mean time loss between sexes.

Specific Comments:

Abstract:

Your title and objective in the abstract do not match with regards to your main objectives?

Please include the stats into your outcome measures

In the results, fever should not be included in the upper/lower cluster.

Conclusions, "Flu-like symptoms dominated." this is not a full sentence, and you will see my comments to come regarding the term "flu-like".

Response: The title has been revised to better align with the study objectives, and now reads "Illness prevalence and symptoms in youth floorball players – a one-season prospective cohort study involving 471 players." We have rewritten the Abstract to incorporate the above suggestions, and to include information about time loss. In the interest of brevity and in order to keep to the word count we have not included more statistics in the Abstract.

Overall comment: I would recommend you have an English editor read through your work, as there are many grammar errors. I have not listed them.

Response: The manuscript has now gone through professional language revision.

Introduction:

There is no mention of floorball in the introduction. There needs to be some context of floorball, and the age group of these athletes (some were only 12 years old), and not elite. Is floorball indoor or outdoor? Is it winter or summer? If it's indoors, is there ventilation, is it on ice? These environmental conditions should be explained to the reader as, I had not heard of floorball before this paper.

Response: Thank you for this important contextual remark. It is mainly a winter indoor sport, the season runs from autumn (September) to Spring (March) as shown in Fig 1 in the manuscript. We have added the following text early in the "Methods" where we believe it fits well, lines 101-103: "Floorball is a winter sport played indoors on a playing surface made of rubber or parquet, and with a plastic stick and light plastic ball on a 40 x 20 m sized pitch. There are five field players and one goalkeeper on the court in each team with free substitutes."

Methods:

Please include a statement regarding informed consent and assent from the parents and children (I assume this was done seeing as this trial was such a large one and you have ethics), but please make sure you explicitly write this.

Response: Thank you for this important comment, and yes of course, informed consent was retrieved. The following sentence has been added to the "Methods", lines 95-96: "Written informed consent was collected from all participating players, and from legal guardians for players below 15 years of age."

Seeing as an athlete could not report an illness if an injury was reported, could you please indicate the number of weeks this was the case (eg. in 200 player weeks an injury was reported and therefore an illness could not be reported due to methodological errors)?

Response: Injury was reported in 948 weeks, we have added this information to the manuscript on lines 145-146.

Could you please explain why you did not use illness incidence? per player days... or player weeks? This is what is recommended in the consensus statement.

Also, considering that many players had multiple illnesses, consider reporting whether the illnesses were subsequent illness, recurrent or subsequent local illnesses. This would be appropriate seeing as you are attempting to follow the consensus guidelines.

Response: We agree that more detailed information about illness characteristics in terms of e.g. new/recurrent/subsequent illness, and illness incidence would have been valuable. However, since we rely on youth athlete self-report of illness data we can only report occurrence and symptoms of illness per weekly report (prevalence). Unfortunately, we don't have any medical verification of reported illnesses and it is impossible to confirm whether e.g. illness reporting over two weeks relate to the same illness or different illness episodes, recurrence or subsequent illness. Hence, we cannot report illness incidence as this only includes new illness cases. We therefore report illness prevalence (season prevalence and weekly prevalence) as this includes both new and ongoing cases, and this is also coherent with the 2020 IOC consensus statement, where illness risk is recommended to be reported as either incidence or period prevalence.

Statistical Analysis:

What was your level of significance?

Response: Thank you for noticing. The significance level was set to $p < 0.05$, this has been added on line 137.

An assumption of the Chi-square test is that you compare frequencies and not percentages? What did you do, seeing you write that you used proportions? Please check this.

Response: Indeed, we have used the Chi-square test to compare frequency distribution between sexes, we have expressed this as both a frequency and proportion (% of all reported symptoms) in Table 2 for easy interpretation (we have switched the order of n (%) in the revised Table 2 to highlight that we compare frequencies). We have corrected the text in the "Statistics" and also included information about comparison of mean days absence per illness week between sexes, as per the reviewer's suggestion to add more results on time loss, lines 134-136.

Results:

Line 139-142: as mentioned in the earlier comments, here you give some information regarding the time-loss data, it would be of interest to the clinician and researcher to give more detail on these data. Include the types of illnesses which result in this, are females more or less affected than males? Is this different during the season?

Response: In line with the reviewer's suggestion we have extended and revised Table 2 and added information about illness impact on participation in sport (full participation, reduced participation, could not participate, as per OSTRC questionnaire question 1) for each reported symptom and symptom cluster. We have also added the mean days absence per illness week total and for each sex on lines 150-151. There was no difference in mean time loss between

sexes.

Illness symptoms: do you have the duration of symptoms? this could add more value than just what symptoms each player had?

Response: We have extended and revised Table 2 and added information about illness impact on participation in sport (full participation, reduced participation, could not participate, as per OSTRC questionnaire question 1) for each reported symptom and symptom cluster. We believe this is the most relevant and most valid data to report related to symptoms. We don't have any data related to duration of symptoms, unfortunately.

Table 2: You appear to have compared the sexes (you report a p-value in the narrative), the p-values should be included in this table, otherwise it appears as if you are hiding data.

Response: P-values have been added to Table 2 for frequency comparisons with the chi-square test.

Please check your % of non-specific illnesses for all.

Response: Thank you for noticing this typo, overall N and % have been re-checked in the whole Table 2.

Swollen glands: this only belongs in upper/lower if they are neck glands, do you have in the information to prove they were neck glands? if not they should be removed.

Response: Based on the reviewer's suggestion we have moved swollen glands into category Other – unspecified, as we have no information about the location of swollen glands, and multiple sites are possible.

Line 172: You have not classified these as flu-like, you shouldn't be writing the narrative like this then. Same goes for figure 2.

Response: We have re-written this throughout the manuscript and including Figure 2.

Discussion:

You list the key findings, however you do not discuss finding 2. Please add this into your discussion.

Response: We have added a discussion about impact of illness on sport participation and time loss on lines 219-224.

Line 184: again you include fever in upper/lower cluster, this goes against your original groupings.

Response: We have corrected the wording in this sentence to clarify that fever is not included in the upper/lower cluster.

Line 197: again, you mention 1 in 2 illnesses lead to time-loss, this needs to be further investigated in the results please.

Response: We have extended the discussion relating to time-loss from sports, see lines 219-224.

Line 201: Please rather use the terminology of the virus (SARS-CoV-2), not the "Novel Coronavirus pandemic"

Response: We have reworded and it now reads "...current COVID-19 pandemic" as per WHO description.

Line 201-206: Please revise this sentence, it is much too long.

Response: The sentence has been split into two sentences.

Line 208: "medial"?

Response: Typo, this should be medical, thank you.

Line 211-228: Here you have extrapolated your data beyond the scope of your article. You have performed no PCR testing to determine that your illnesses were influenza. Vaccines are beyond the scope of this article.

Response: We have omitted most of this section as we agree it is speculative and partly outside the scope of the study. We have kept a shorter section with the observation that the peak in illness prevalence, and fever and respiratory symptoms, seen in our youth floorball cohort coincides with a peak in influenza B in the Swedish general population of young people in the same period. We believe this is noteworthy observation and indication that influenza is a likely cause of illness also in our floorball cohort at this time of the year.

Comparison between sexes: Please remember the average age of your population was 13, at this age females are more mature than males, so the differences between males and females would be smaller. This paragraph reads as though you investigated elite adults.

Response: Thank you, we have omitted parts of this section in line with your comment as it was somewhat speculative.

Line 242-244: I am very confused as to what you are trying to say here, but again this seems to apply to adult females anyway, and not 13 year olds.

Response: The sentence has been rewritten to clarify that we don't have any data to support reasons for the observed sex difference here.

Line 247: please include that these were 13 year old children answering a medical questionnaire, there is a chance that they did not fully understand the illness questions.

Response: This has been added as suggested, lines 238-239.

Line 251-252: This is not a very high response rate, there were 26 weeks, so please include that overall this was a low response rate (58%).

Response: We have added that the overall response rate was low (response rate was 61%), although in line with several similar studies with self-report of health problems in youth sports, e.g. 60-66% in studies by Moseid et al. SJMSS 2018 (ref 8) and von Rosen et al. JAT 2018 (new reference added), see lines 242-244.

Conclusions:

Please remove the flu-like sentence. And look at the repetition of "risk mitigation".

Response: The conclusion has been re-written as suggested.

VERSION 2 – REVIEW

REVIEWER	Philippe Tscholl F-MARC
REVIEW RETURNED	25-Oct-2021

GENERAL COMMENTS	I have no more comments to add
--------------------------------

REVIEWER	Nicola Sewry University of Cape Town, Human Biology
REVIEW RETURNED	02-Nov-2021

GENERAL COMMENTS	Dear Authors, Thank you very much for your comprehensive revision, you have addressed all my concerns. I think the paper is going to make a great contribution to literature. I have a few minor edits for your consideration. Please remove the word "arranged" from the abstract (under outcome measures). Line 87: please replace illnesses with illness Line 103: consider adding that each team is allowed a maximum of 4 substitutes on the bench. Line 166: I wonder whether you can say that the fainting was truly more in the female players, seeing as you could not perform stats on this? I will leave this up to your discretion. Line 248: This sentence is unclear, what do you mean by "response burden"?
--

VERSION 2 – AUTHOR RESPONSE

1. Please remove the word "arranged" from the abstract (under outcome measures).

Reply: Changed accordingly

2. Line 87: please replace illnesses with illness

Reply: Changed accordingly

3. Line 103: consider adding that each team is allowed a maximum of 4 substitutes on the bench.

Reply: This does not apply to youth floorball players in Sweden, where a free number of players are allowed as substitutes on the bench. No changes made.

4. Line 166: I wonder whether you can say that the fainting was truly more in the female players, seeing as you could not perform stats on this? I will leave this up to your discretion.

Reply: We argue that 0 cases among males and 20 among females is a noteworthy difference even though the Chi square test cannot be applied. No changes made.

5. Line 248: This sentence is unclear, what do you mean by "response burden"?

Reply: We see that this sentence serves no purpose as it stands and have therefore removed it. We also slightly re-arranged the section to improve the flow of the text.